# Synthesis and Characterization of Ceria- and Samaria-Based Powders and Solid Electrolytes as Promising Components of Solid Oxide Fuel Cells

**Marina V. Kalinina [1], Daria A. Dyuskina [1], Maxim Y. Arsent'ev [1], Sergey V. Mjakin [2,3,\*] and Olga A. Shilova [1,2,4]**

[1] Institute of Silicate Chemistry of the Russian Academy of Sciences, 2 Makarova Emb, 199034 St. Petersburg, Russia

[2] Department of Theory of Materials Science, Saint-Petersburg State Institute of Technology, Technical University, 26 Moskovsky Prospect, 190013 St. Petersburg, Russia

[3] Institute for Analytical Instrumentation, Russian Academy of Sciences, 31-33A Ivana Chernykh Str., 198095 St. Petersburg, Russia

[4] Department of Nanotechnology and Nanomaterials for Radioelectronics, Saint-Petersburg State Electrotechnical University "LETI", 5 Professora Popova Str., 197376 St. Petersburg, Russia

[\*] Correspondence: svmjakin@technolog.edu.ru

**Abstract:** Finely dispersed $(CeO_2)_{0.95}(Sm_2O_3)_{0.05}$, $(CeO_2)_{0.90}(Sm_2O_3)_{0.10}$ and $(CeO_2)_{0.80}(Sm_2O_3)_{0.20}$ mesoporous powders with a specific pore volume of $0.080–0.092$ $cm^3/g$ and a specific surface of $50–83$ $m^2/g$ are synthesized by the co-precipitation of cerium and samarium hydroxides from the corresponding nitrate solutions. The prepared powders are used to obtain ceramic nanomaterials with a fluorite-like cubic crystal lattice with a coherent scattering region (CSR) of about $65–69$ nm ($1300$ °C). The study of physicochemical and electrophysical properties of the prepared ceramics revealed the obtained materials featuring an open porosity of $2–6\%$ and a predominantly ionic type of electric conductivity (ion transport numbers $t_i = 0.85–0.73$ in the temperature range $300–700$ °C). The conductivity in solid solutions proceeds via a vacancy mechanism with $\sigma_{700\,°C} = 3.3\cdot10^{-2}$ S/cm. The synthesized ceramic materials are shown to be promising as solid oxide electrolytes in medium temperature fuel cells.

**Keywords:** co-precipitation of hydroxides; fuel cells; ceramic electrolytes; nanomaterials

## 1. Introduction

Most of the power currently produced worldwide is generated using fossil hydrocarbons (coal, fuel oil, natural gas) as fuel, which adversely affects the environment. Presently, the international community is deeply concerned about the problem of global warming. In this regard, the attention of the world's scientific community is focused on the search for alternative energy sources, particularly including the development of hydrogen energy. The use of hydrogen and fuel cells, which are the basis of the so-called hydrogen-oriented economy, opens up a completely unique approach to obtaining "safe" energy that has no impact on the climate, provides an increase in energy efficiency and contributes to the development of energy sources free from greenhouse gas emissions [1].

One of the ways to convert fuel energy into electricity with high efficiency (up to 70–80%) is through using solid oxide fuel cells (SOFCs), in which the chemical energy of the fuel is directly converted into electrical energy during oxidation [2–4].

Similar to all types of fuel cells, SOFCs involve a cathode and anode being separated by an electrolyte, in this case comprising a ceramic electrolyte responsible for the transfer of oxygen ions. The development of SOFC-based commercial power generation systems is becoming a primary objective for distributed power generation, energy conservation, cogeneration and saving fuel resources [1].

Furthermore, SOFCs are highly promising due to their enhanced resistance to fuel contamination and versatility of applicable combustible gases in comparison with other FC types [5–7]. The applied electrolyte must have a unique set of physicochemical and mechanical properties including highly specific electrical conductivity, optimal parts of ionic conductivity, high strength and low porosity.

A certain disadvantage of SOFCs relates to high working temperatures contributing to their high costs and problems of compatibility between the electrolyte and electrode materials [2,7]. To increase the economic competitiveness of SOFCs, their operating temperature should be reduced to 600–700 °C, in tandem with maintaining high ionic conductivity of the applied solid electrolytes.

Presently, solid electrolytes for SOFCs working in medium temperature region (400–800 °C) often comprise cerium oxide doped with rare earth oxides [8,9]. Ceria-based electrolytes possess an advantageous combination of high unipolar oxygen conductivity, chemical stability in carbon- and hydrogen-containing atmospheres and an inertness to electrode materials, which makes them highly promising for medium-temperature fuel cells.

Being comparable with conventionally used zirconia with respect to electric performances, $CeO_2$-based electrolyte nanomaterials allow for a significant reduction of working temperature for fuel cells, thus providing an increased durability and simplified structure [9,10]. Moreover, these materials have almost an order of magnitude higher ionic conductivity than solid solutions based on $ZrO_2$ because of a large radius of $Ce^{4+}$ ions providing the formation of a more "open" crystal structure for the migration of $O^{2-}$ ions [4]. In this context, a promising approach is the application of a $CeO_2$-$Sm_2O_3$ system featuring the highest dopant solubility in tandem with maintaining high ionic conductivity and usefulness for obtaining solid electrolytes maintaining a high stability of electrophysical performances within long time exploration, including high temperature gradients and fluctuations [4,7]. According to the $CeO_2$–$Sm_2O_3$ phase diagram, the concentration range of 0–20 mol.% $Sm_2O_3$ is mostly interesting for studies of this system since it is a single-phase region containing only fluorite-like cubic solid solutions [4,8].

The studies relating to the preparation of nanostructured oxide materials and characterization of their properties are of great interest in the field of materials science. In this regard, research on the development of methods for obtaining ultrafine powders and, on their basis, new types of materials with target properties takes on particular relevance for creating a new generation of ceramic nanomaterials for modern technology [9,11,12]. In particular, the preparation of bulk solid electrolytes with high exploration performances requires the application of finely dispersed powders [10]. The most preferable approaches to such powders are liquid-phase methods, including hydrothermal synthesis, the sol-gel method, the Pechini method, co-precipitation of hydroxides from solutions of inorganic salts and co-crystallization of salts. Co-precipitation of hydroxides with a low-temperature treatment is of particular significance, since it provides a more precise adjustment of the target products' dispersion and microstructures by the variation of the synthesis conditions, which affords single-phase weakly agglomerated xerogels and nanosized powders of a given composition with high specific surface areas [11–14]. This method provides great uniformity of the resulting materials on the atomic layer, and more precise stoichiometry and reduction of the powder synthesis temperature by 300–600 °C compared with such conventional processes such as solid phase synthesis [15]. This allows for a reproducible preparation of products with the required composition and dispersion (5–100 nm) via a quite simple procedure free of expensive equipment [7,16]. The uniformity of resulting materials strongly depends on various technological factors including the composition and concentrations of applied reactants and precipitating agents, sequence of mixing, process temperature and aging conditions (precipitate residence time in the mother liquor) [16,17]. In order to improve the dispersion of the target powders (due to the suppression of the precipitates particles coagulation and prevention from their agglomeration), the precipitation should be performed using diluted solutions (~0.1 M) [7]. In addition, a highly effective approach to avoid coagulation is a low temperature processing of co-precipitated

hydroxides [16]. Low-temperature treatment of the precipitate makes it possible to weaken the forces of interaction between the crystalline particles of the precipitate, to carry out deep dehydration of the gels and to maintain the high dispersity and chemical homogeneity of the co-precipitated powders. However, it should be taken into account that rapid freezing of precipitates using liquid nitrogen temperatures can contribute to incomplete cryocrystallization of hydroxides with the preservation of a certain amount of the amorphous phase in cryogranules. This will contribute to segregation of individual components and violation of the homogeneity of the precipitation product. Therefore, in some cases, it is advisable to freeze the precipitated product at temperatures from $-50$ to $-25\ °C$ [11,13]. It is desirable to carry out low-temperature treatment of the precipitated product at temperatures from $-30$ to $-25\ °C$ to achieve a complete cryocrystallization of hydroxides without disturbing the homogeneity of the precipitated product.

The aim of this research is a directed liquid-phase synthesis by co-precipitation of hydroxides with low-temperature treatment and characterization of physicochemical and electrophysical properties of nanodispersed powders in the $CeO_2$-$Sm_2O_3$ system and ceramic electrolyte materials based on solid oxide fuel cells.

## 2. Experimental

*2.1. Synthesis of Xerogels, Powders and Ceramic Materials of the Composition $(CeO_2)_{1-x}(Sm_2O_3)_x$ (x = 0.05, 0.10, 0.20)*

The synthesis of xerogels and powders with different concentration ratios of oxides in the $CeO_2$-$Sm_2O_3$ system was carried out by co-precipitation of cerium and samarium hydroxides followed by low temperature processing.

For the synthesis, nitric acid salts of cerium $Ce(NO_3)_3·6H_2O$ (analytical purity grade with the reagent content higher than 98% wt.) and samarium $Sm(NO_3)_3·6H_2O$ (chemical purity grade with the reagent content higher than 99% wt.) were used, from which diluted ($\sim$0.1 M) solutions were prepared. A 1M aqueous solution of ammonia hydrate ($NH_3·H_2O$) was used as a precipitating agent.

Taking into account the pH values required for the precipitation of each hydroxide (preliminarily determined by potentiometric titration using a 150M pH meter (Teplopribor, Russia)), the pH value of the reaction mixture was maintained in the range 10–11 at $NH_3·H_2O$ concentration of about 1M. The synthesis was carried out using a reverse co-precipitation method with a minimum rate ($V_{prec}$ = 0.02 $cm^3$/s) at thorough stirring. The resulting gel-like hydroxide precipitate was filtered followed by freezing at $-25\ °C$ within 24 h to provide deagglomeration and maintain a high dispersion of the co-precipitated product in the $CeO_2$-$Sm_2O_3$ system. Freezing of the gel allows for the removal of adsorption and crystallization water from the precipitate coupled with rapid hardening, thus providing high chemical homogeneity in the solid phase. The applied low-temperature treatment during the synthesis process affords microstructure evolution and makes it possible to obtain more finely dispersed products [11]. X-ray amorphous xerogels obtained as a result of drying (150 °C, 1 h) were subjected to heat treatment (600 °C, 1 h) to form a stable crystalline structure of powders. Then, the obtained powders were consolidated by uniaxial cold pressing at 150 MPa, followed by sintering at 1300 °C within 2 h.

### 2.2. Material and Methods

X-ray diffraction analysis (XRD) was performed using a D8-Advanse diffractometer (Bruker, Billerica, MA, USA). The international database ICDD-2006 was used to interpret the diffraction patterns; the analysis results were processed using the WINFIT 1.2.1 program using the Fourier transform of the reflex profile. The size of coherent scattering regions (CSR) was estimated using PDWin software based on the Williamson–Hall method, taking into account broadening of the peaks due to both small particle sizes and micro-tensions in the crystals. In this study, micro-tensions were negligibly small for all the samples.

The samples' structure was also characterized by SEM measurements using a Tescan Vega 3 installation.

Differential thermal analysis (DTA) was used to study thermolysis processes occurring in co-precipitated xerogels and powders when heated in the temperature range of 20–1000 °C (Gerivatograph Q-1000 produced by MOM).

The specific surface area of the synthesized powders was measured by low-temperature nitrogen adsorption using a QuantaChrome Nova 4200B analyzer. Based on the data obtained, the specific surface area SBET of the samples was calculated using the Brunauer–Emmett–Teller (BET) model.

The calculation of the pore size distribution was carried out on the basis of nitrogen desorption isotherms according to the Barrett-Joyner-Halenda (BJH) method. Thermal treatment of the powders was carried out in a Naberterm furnace with program control in the temperature range of 25–1300 °C for 1–2 h, followed by slow cooling of the furnace. The open porosity of the samples was determined by hydrostatic weighing in distilled water in accordance with Russian Standard GOST 473.4-81 [18].

The electrical resistance of the obtained ceramic materials was measured by a two-contact method using a direct current in the temperature range of 250–1000 °C using the "Hardware-software installation for investigating the electrical properties of nanoceramics in different gas media" [12]. The transfer numbers of ions and electrons in bulky solid electrolytes were determined by the West–Tallan method [19].

A $CO_2$-CO mixture was used as an inert gas (corresponding to oxygen partial pressure of 1000 Pa). The measurements were carried out using a direct current in weak (U = 0.5 V) fields after a long (up to 30 min) drop of the current. The contributions of ionic and electronic conductivity were estimated as:

$$t_e = R_{air}/R_e \tag{1}$$

and

$$t_i = 1 - \underline{t_e} \tag{2}$$

where $t_e$ and $t_i$ are the transport numbers of electrons and ions, respectively, $R_{air}$ and $R_e$ are the sample resistance measured in air and in inert gas atmosphere.

### 3. Results and Discussion

#### 3.1. Study of the Synthesized Xerogels Thermolysis

The thermal behavior of the considered xerogels with a given composition obtained by co-precipitation of hydroxides followed by low-temperature treatment was studied using differential thermal analysis with a heating rate of 2 °C/min up to 1000 °C. As an example, Figure 1 shows a thermogram of a xerogel with the composition $(CeO_2)_{0.90}(Sm_2O_3)_{0.10}$, obtained by co-precipitation without (Figure 1a) and with a low-temperature treatment (Figure 1b).

According to Figure 1a, the endothermic effect with the peak at ~110 °C is due to the removal of physically bound water from the surface of xerogel particles. The dehydration of crystalline hydrate, as well as the decomposition of nitric acid salts, proceeding in one stage, correspond to a weight loss of ~31%. In the range of 260–280 °C, a relatively narrow exothermic effect is observed for these powders, which may be determined by the crystallization of a cubic solid solution of the fluorite-type based on cerium oxide. This temperature range is close to the value of about 310 °C observed for a similar composition $(CeO_2)_{0.80}(Sm_2O_3)_{0.20}$ reported in [20].

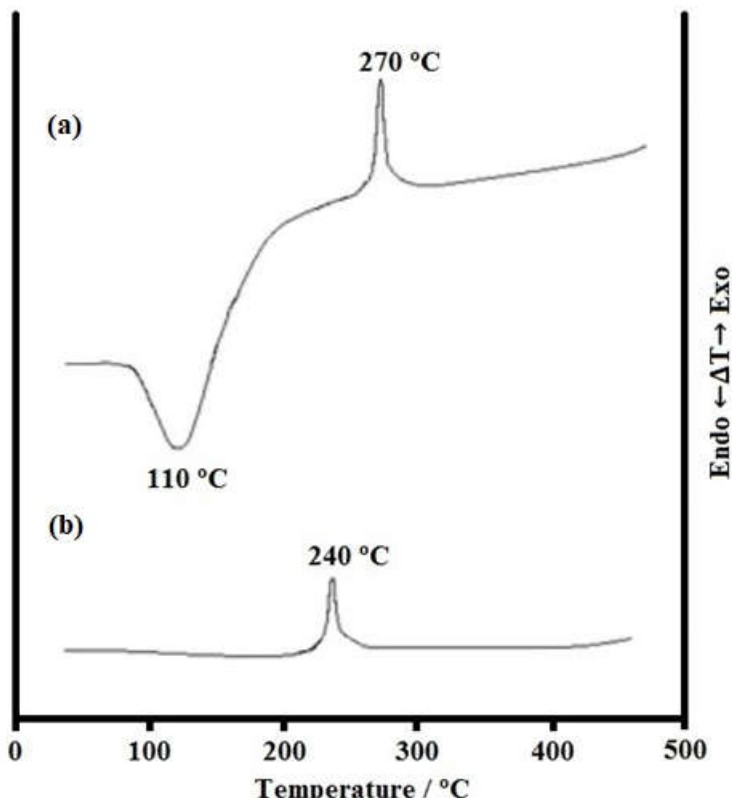

**Figure 1.** Differential thermal analysis results for $(CeO_2)_{0.90}(Sm_2O_3)_{0.10}$ xerogel prepared without (**a**) and with (**b**) xerogel freezing at $-25\,°C$ (24 h).

As can be seen from Figure 1b, in the crystalline hydrate obtained with a low-temperature treatment, no endothermic effect corresponding to the dehydration process is observed. These results indicate that most of the water is removed during low temperature treatment. When low-temperature xerogel treatment is used, the crystallization temperatures of the solid solution based on cerium oxide are reduced from 260–280 °C to about 230–250 °C. The weight loss of the crystalline hydrate was about 9%.

*3.2. Low-Temperature Nitrogen Adsorption Characterization of the Microstructure of Powders Synthesized by Co-Precipitation of Hydroxides*

As an example, Figure 2 shows an adsorption–desorption isotherm and a differential pore size distribution for $(CeO_2)_{0.90}(Sm_2O_3)_{0.10}$ powder prepared with heat treatment at 600 °C. The adsorption–desorption isotherm shown in Figure 2 corresponds to type IV according to the IUPAC classification, corresponding to the mesoporous structure. The specific profile of capillary-condensation hysteresis suggests that slit-shaped pores (type H3 according to the IUPAC classification) predominate in this powder. For this sample, the total pore volume is 0.086 cm$^3$/g and the specific surface area is 78 m$^2$/g, with small mesopores (1–10 nm) generally prevailing (Figure 2).

The characteristics of the microstructure for powders synthesized by co-precipitation of hydroxides are summarized in Table 1. The obtained results show that with an increase in the $Sm_2O_3$ concentration, the specific pore volume and specific surface area of the studied powders increase in the range of 0.080–0.092 cm$^3$/g and 50–83 m$^2$/g, respectively, and the average pore size decreases from 3.6 to 1.5 nm. The specific surface depends not only on the powder particle size, but also upon the degree of the surface development (roughness) and specific pore volume. The observed pore size decreases in tandem with the growth of the pore volume and the specific surface probably due to the distortion of chemical bond systems and the formation of additional element-oxygen bonds at the particles formation, resulting in the increase of their package density.

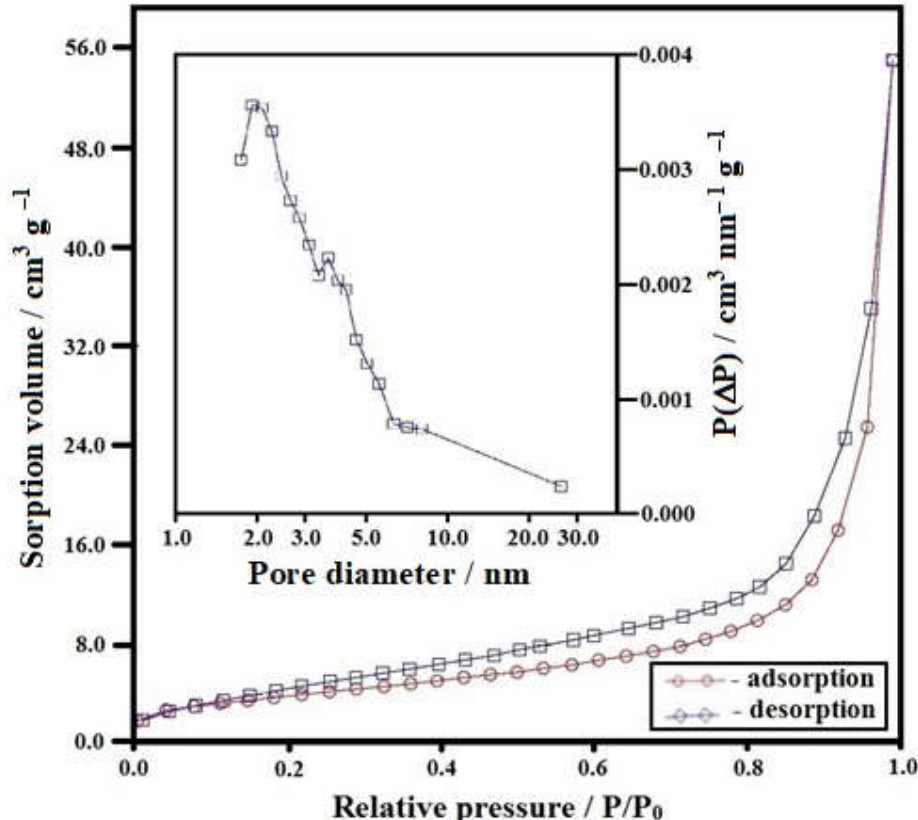

**Figure 2.** Adsorption–desorption isotherm and differential pore size distribution (caption) for $(CeO_2)_{0.90}(Sm_2O_3)_{0.10}$ powder (annealing at 600 °C).

**Table 1.** Microstructure characteristics of the powders synthesized by co-precipitation of hydroxides, determined by low-temperature nitrogen adsorption.

| Composition | Specific Surface Area $S_s$, $m^2/g$ | Average Pore Size $D_{por}$, Nm | Specific Pore Volume $V_{por}$, $cm^3/g$ |
|---|---|---|---|
| $(CeO_2)_{0.95}(Sm_2O_3)_{0.05}$ | 50 | 3.6 | 0.080 |
| $(CeO_2)_{0.90}(Sm_2O_3)_{0.10}$ | 78 | 2.5 | 0.086 |
| $(CeO_2)_{0.80}(Sm_2O_3)_{0.20}$ | 83 | 1.5 | 0.092 |

*3.3. Study of the Crystal Structure of the Obtained Powders*

The results of XRD characterization of the obtained powders crystal structure are in good agreement with the data of low-temperature nitrogen adsorption. Based on XRD results, it can be concluded that annealing at 600 °C for 1 h results in highly dispersed solid solutions with a cubic structure of the fluorite type with an average CSR size of ~7 nm. As an example, the fluorite type cubic solid solution formation sequence for a sample with $(CeO_2)_{0.90}(Sm_2O_3)_{0.10}$ composition is shown in Figure 3. During further annealing at 600–1300 °C, the single-phase nature of the ceramic samples obtained from the synthesized powders remains.

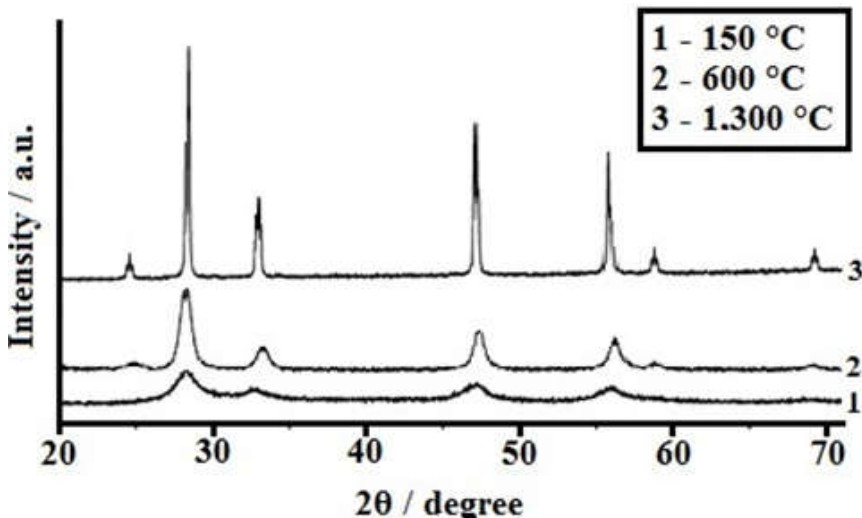

**Figure 3.** XRD profiles for $(CeO_2)_{0.90}(Sm_2O_3)_{0.10}$ (a = 5.4391 Å)-based xerogel (1–150 °C), powder (2–600 °C) and ceramics (3–1300 °C) samples synthesized by co-precipitation of hydroxides followed by low-temperature processing.

SEM images indicate that the powder comprises large (more than 100–150 μm) agglomerates and a small amount of separate 1–10 μm sized particles (Figure 4a), whereas the compressed pellets annealed at 1300 °C consist of particles with an average size of about 500 nm, probably involving several crystallites (about 65–70 nm according to XRD data) tightly adjacent to each other without prominent boundaries (Figure 4b).

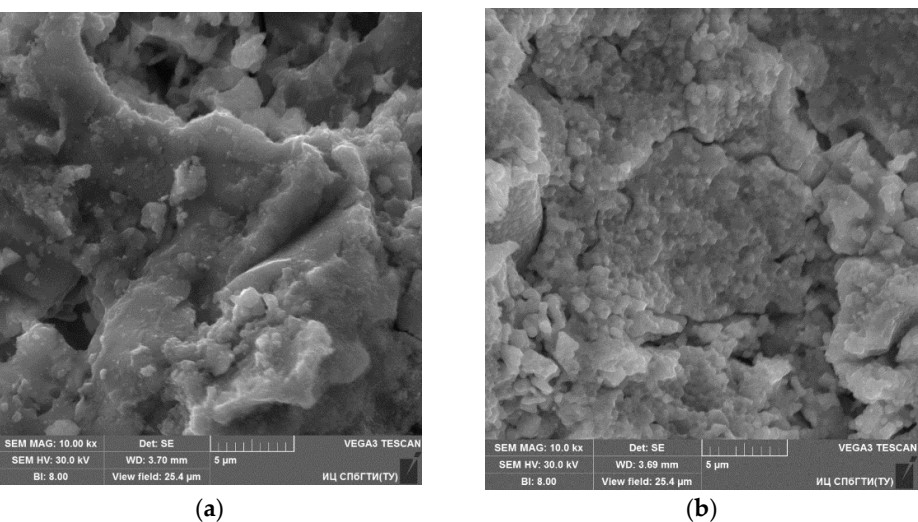

**Figure 4.** SEM images of (**a**) initial powder (annealed at 600 °C) and (**b**) ceramics (annealed at 1300 °C).

Physicochemical properties of all the ceramic samples obtained on the basis of powders in the $CeO_2$-$Sm_2O_3$ system, synthesized by co-deposition methods followed by low-temperature treatment, are summarized in Table 2. These data show that an increase in the content of samarium oxide in the obtained samples leads to a decrease in their density, which is probably due to the distortion of the cerium dioxide lattice upon $Sm_2O_3$ dissolving. In addition, a slowdown in the growth of crystallites is observed. At the same time, in the case of $(CeO_2)_{0.90}(Sm_2O_3)_{0.10}$ based ceramics, an increase in relative density and open porosity is observed in comparison with the previously studied $(CeO_2)_{0.90}(Y_2O_3)_{0.10}$ solid solution. Generally, it can be concluded that the applied co-precipitation method makes it possible to obtain dense and highly dispersed samples, which is consistent with the results obtained earlier [11].

**Table 2.** Physicochemical properties of $(CeO_2)_{1-x}(Sm_2O_3)_x$ (x = 0.05; 0.10; 0.20) ceramics samples synthesized by co-precipitation.

| Composition | [b] $\rho_{teor}$ g/cm$^3$ | [a] Apparent Density $\rho_{exp}$ g/cm$^3$ | [c] $\rho_{rel}$, % | CSR, nm (1300 °C) | [f] P, % | [d] $\sigma_i \cdot 10^{-2}$ S/cm (700 °C) | [e] $E_a$, eV | Structure, Parameter a |
|---|---|---|---|---|---|---|---|---|
| $(CeO_2)_{0.95}(Sm_2O_3)_{0.05}$ | 7.23 | 6.55 | 91 | 69 | 2.0 | 1.2 | 1.35 | F a = 5.4292 |
| $(CeO_2)_{0.90}(Sm_2O_3)_{0.10}$ | 6.98 | 6.33 | 91 | 68 | 3.8 | 2.7 | 1.31 | F a = 5.4391 |
| $(CeO_2)_{0.80}(Sm_2O_3)_{0.20}$ | 6.90 | 6.25 | 91 | 65 | 6.2 | 3.3 | 1.29 | F a = 5.4471 |

Notes: (a) $\rho_{exp}$—experimentally measured density. (b) $\rho_{teor}$—theoretical density. (c) $\rho_{rel}$—relative density. (d) $\sigma_i$—ionic conductivity. (e) $E_a$—activation energy. (f) P—open porosity.

### 3.4. Electrical Properties of Ceramic Samples

The electrical conductivity of $(CeO_2)_{1-x}(Sm_2O_3)_x$ samples (x = 0.05; 0.10; 0.20) was measured using the two-contact method with a direct current (Figures 5 and 6). The appearance of high oxygen ionic conductivity in $CeO_2$-$Sm_2O_3$-based solid electrolytes is determined by the formation of oxygen vacancies in the $CeO_2$ matrix when $Ce^{4+}$ is replaced by $Sm^{3+}$ during the dissolution of $Sm_2O_3$ in $CeO_2$, which can be described by the following quasi-chemical equation in the Kroeger–Winke notation [17]:

$$Sm_2O_3 \xrightarrow{CeO_2} 2Sm'_{Ce} + 3O_O^X + V_O^{\cdot\cdot} \tag{3}$$

where $Sm'_{Ce}$ is a samarium ion replacing $Ce^{4+}$ and yielding a negative charge, $V_O^{\cdot\cdot}$ is a positively charged oxygen vacancy compensating the dopant charge, $O_O{}^X$ is oxygen atom in a regular site with a neutral charge.

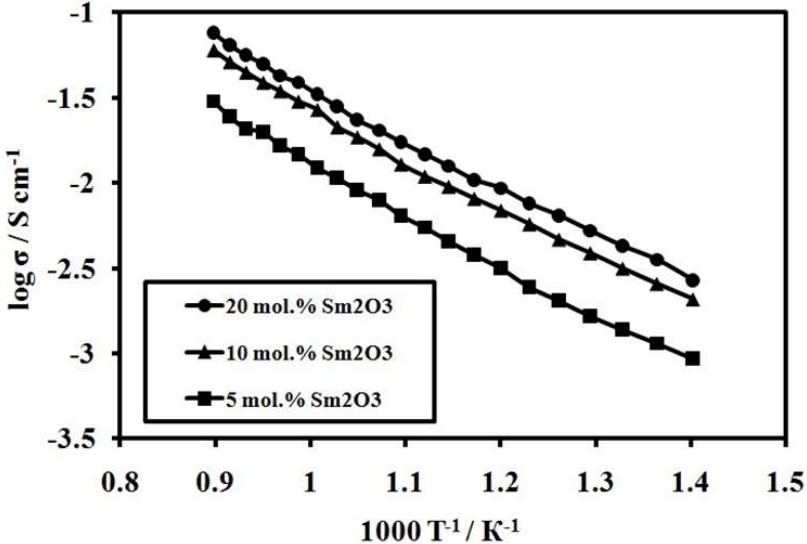

**Figure 5.** Temperature dependence for specific electrical conductivity of $(CeO_2)_{0.95}(Sm_2O_3)_{0.05}$ (■); $(CeO_2)_{0.90}(Sm_2O_3)_{0.10}$ (▲) and $(CeO_2)_{0.80}(Sm_2O_3)_{0.20}$ (●) ceramics samples.

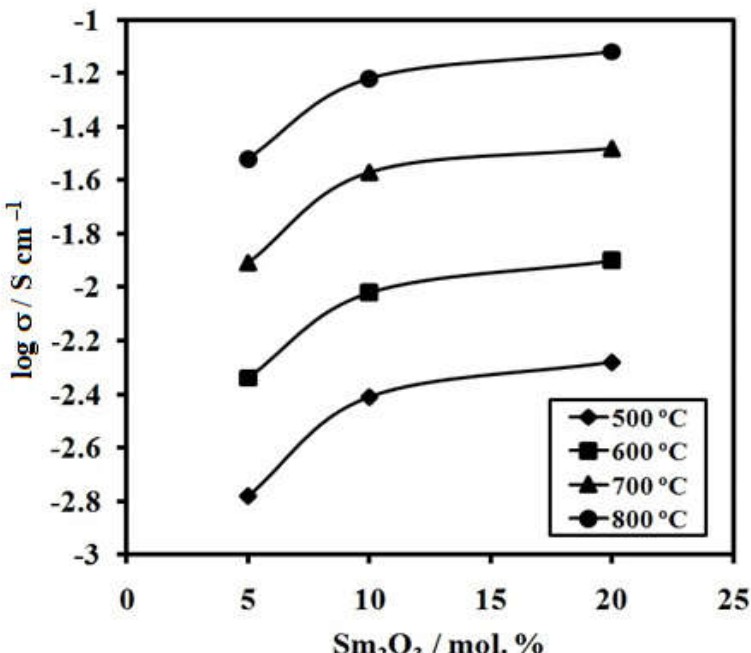

**Figure 6.** Specific electrical conductivity of ceramics samples at temperatures 500 °C (◆), 600 °C (■), 700 °C (▲) and 800 °C (●) as function of $Sm_2O_3$ concentration.

As shown in Figure 5, the temperature growth in the range from 500 to 1000 °C leads to the increase in electrical conductivity of all the samples. In addition, with an increase in the concentration of samarium oxide, the specific electrical conductivity of the ceramics increases in the entire temperature range in study (Figure 6). The highest specific electrical conductivity in the temperature range of 500–1000 °C ($\sigma_{700\,°C} = 3.3 \times 10^{-2}$ S cm$^{-1}$) is observed for the sample containing 20 mol.% $Sm_2O_3$. Using the West–Tallan method, the ratio of the electronic and ionic conductivity in the studied ceramic samples was determined.

As an example, Table 3 presents the data on the ratio of the transfer numbers of ions and electrons for the studied samples of the composition $(CeO_2)_{0.90}(Sm_2O_3)_{0.10}$. These data indicate that these solid electrolytes have mixed conductivity with the ion transport number $t_i = 0.85$ at 300 °C and 0.73 at 700 °C. The temperature growth leads to a sharp increase in the contribution of the electronic component to the total value of electrical conductivity that relates to a partial transition $Ce^{4+} \rightarrow Ce^{3+}$.

**Table 3.** Characteristics of mixed electrical conductivity for $(CeO_2)_{0.90}(Sm_2O_3)_{0.10}$ sample: transfer numbers of ions ($t_i$) and electrons ($t_e$) at different temperatures.

| T, °C | $t_i$ | $t_e$ |
|---|---|---|
| 300 | 0.85 | 0.15 |
| 400 | 0.80 | 0.20 |
| 500 | 0.78 | 0.22 |
| 600 | 0.75 | 0.25 |
| 700 | 0.73 | 0.27 |

### 3.5. Ab Initio Studies of Samarium-Doped Ceria

The fluorite structure of $CeO_2$ contains large and empty octahedral sites, which allow for fast oxygen diffusion and make $CeO_2$ an excellent material for the application in solid oxide fuel cells (SOFCs) (Figure 7). The application of aliovalent cation doping can provide oxygen vacancies responsible for oxygen transport. Furthermore, in this case, vacancy-dopant Coulomb interactions emerge significantly, contributing to the activation energy of oxygen vacancy migration [21].

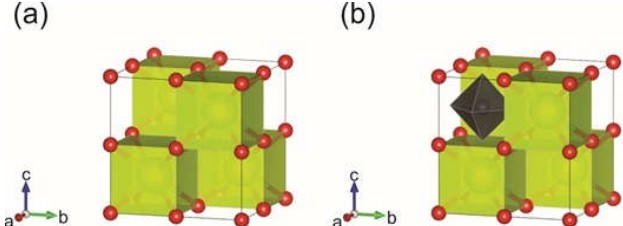

**Figure 7.** The fluorite structure of ceria (**a**) and the fluorite structure of ceria containing the empty octahedral void (grey) (**b**). The light green and red balls represent Ce and O atoms, respectively.

For $Ce_{1-2x}Sm_{2x}O_{2-x}$ ($x$ = 0.05, 0.10, 0.15) solid solutions, the cation sublattice occupation is partial. Nevertheless, the calculation of the properties of this material by the ab initio method requires a cell in which there would be no partial filling of the crystallographic positions. Using the enumeration technique [22] described in the Methods section and DFT calculations, we generated such a cell for $Ce_{0.8}Sm_{0.2}O_{1.9}$ only (Figure 8a). As a result, even after enumerating 1000 variants, we identified only two configurations with different Ewald sums (Figure 8b–d) and similar total energy values (Figure 8e). For the further studies, configuration #1 was chosen.

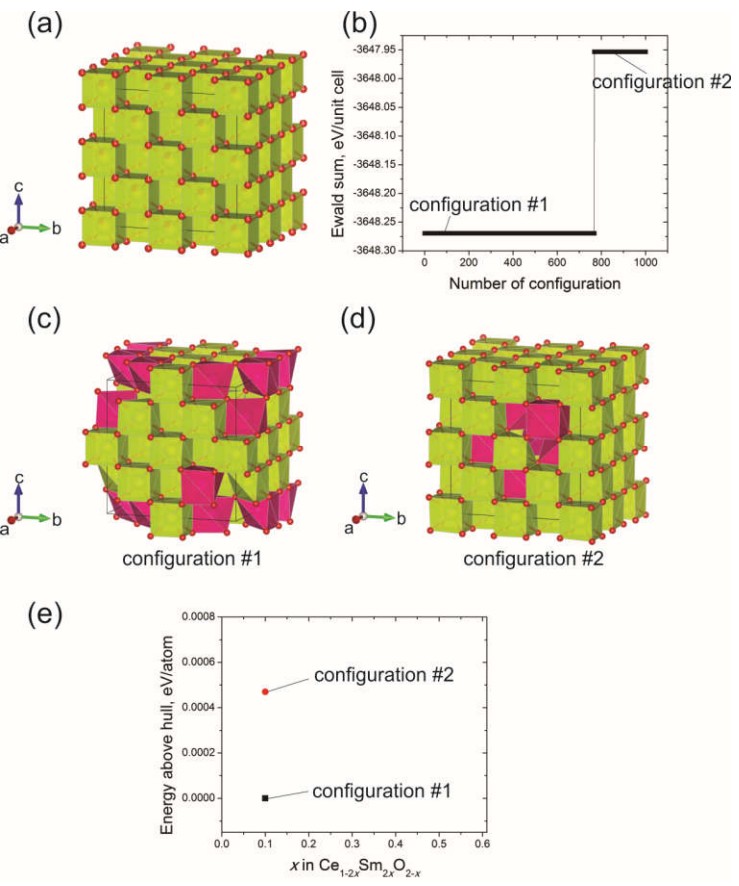

**Figure 8.** (**a**) Structure of $Ce_{0.8}Sm_{0.2}O_{1.9}$: the light green, pink and red balls represent Ce, Sm and O atoms, respectively. 95% occupancy of O (red) is indicated by a partially colored site. Ce sites are partially occupied by Sm (19%); (**b**) Ewald summation results for all possible configurations of $Ce_{0.8}Sm_{0.2}O_{1.9}$ solid solution. View of configurations #1 and #2; (**c**,**d**) Ewald summation according to (**b**); (**e**) Energy above hull values for configurations #1 and #2.

The results obtained using BVS analysis confirm the existence of large empty sites, which allow for fast oxygen diffusion (Figure 9). In this case, large cube-shaped cavities

are present, connected by their vertices through narrow channels (Figure 9b). The value $\Delta BVS = 0.3$ was used in the calculations.

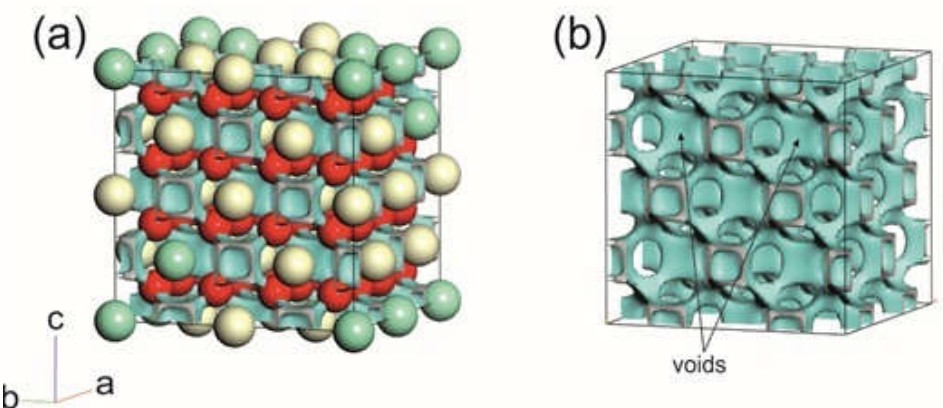

**Figure 9.** Oxygen-ion migration pathways obtained by BVS ($\Delta BVS = 0.3$) for $Ce_{0.8}Sm_{0.2}O_{1.9}$ (turquoise clouds) with (**a**) and without (**b**) crystal structure of $Ce_{0.8}Sm_{0.2}O_{1.9}$.

The BVS method allows for quick preliminary analysis and usually agrees well with experimental data and the results obtained by more accurate methods such as molecular dynamics and density functional theory [23,24]. Despite some advantages of BVS, the method deals with a static model of the crystal structure, not considering possible lattice distortions or atom displacements during ion migration.

Using the DFT-NEB method, nine possible sites for oxygen anion migration in $Ce_{0.8}Sm_{0.2}O_{1.9}$ (configuration #1) were studied (Figure 10a). The migration barrier can be represented as the sum of association energy, resulting from the electrostatic interaction between migrating vacancy $E_{ass}$ and dopant and $E_m$ for bulk diffusion ($\approx 0.8$ eV) (Figure 10b).

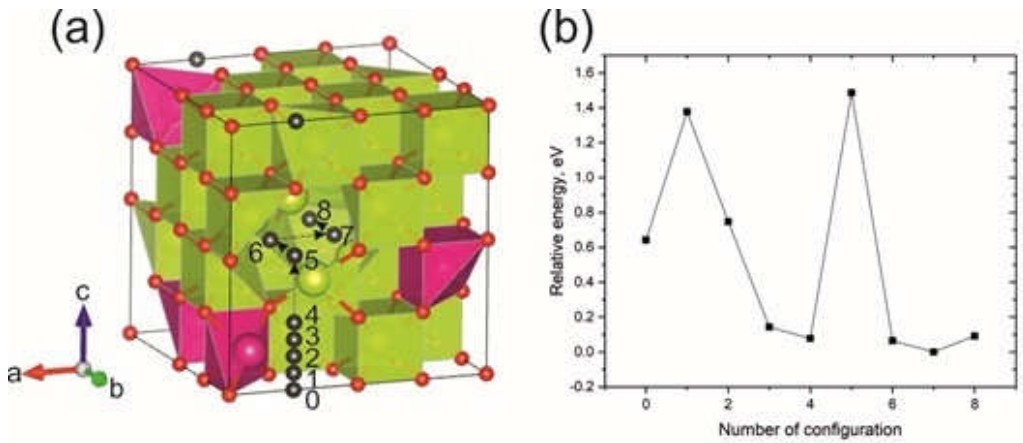

**Figure 10.** (**a**) The numbers designate different locations that one oxygen anion can occupy with respect to the dopants (grey balls). (**b**) Energy profile along 0–8 path. The light green, pink and red balls represent Ce, Sm and O atoms, respectively.

By moving the oxygen anion along the 0–9 path, the value of the total activation energy equal to 1.38 eV can be obtained, which is quite close to the experimental value obtained by us (1.29 eV). The $E_{ass}$ can be evaluated as the energy difference between the nodal sites with the highest and the lowest relative energy (0 and 7). Our calculations show that $E_{ass}$ make up a significant part of the energy barrier. In addition, the relative energy of the oxygen anion in nodal site 5 is high. Site 5 is nodal, unlike another interstitial sites with a high relative energy (site 1), which means that its high energy is most likely due to the electrostatically unfavorable configuration. The oxygen anion can move along an alternative path in which this site is not involved.

## 4. Conclusions

Powders of the composition $(CeO_2)_{1-x}(Sm_2O_3)_x$ (x = 0.05; 0.10; 0.20) with an average crystallite size of ~7 nm, mesoporous structure with pores sizes in the range of 1.5–3.6 nm, total pore volume in the range 0.080–0.092 cm$^3$ g$^{-1}$ and specific surface areas of 50–83 m$^2$ g$^{-1}$ were synthesized by the co-precipitation of hydroxides with low-temperature treatment, followed by obtaining ceramics with a given composition by consolidation of the powders. All the obtained powders and ceramic materials based thereon have a single-phase cubic structure of the fluorite type in the temperature range of 600–1300 °C. The ceramic materials are characterized by a CSR of 65–69 nm (1300 °C), open porosity in the range of 2–6%, a relative density of 91% and ionic conductivity $\sigma_{700\,°C} = 3.3 \times 10^{-2}$ S cm$^{-1}$ and ion transfer numbers $t_i = 0.85$–0.73 in the temperature range of 300–700 °C.

Electrical conductivity in $(CeO_2)_{1-x}(Sm_2O_3)_x$ the solid solutions (x = 0.05; 0.10; 0.20) follows a vacancy mechanism with the specific electrical conductivity values of $\sigma_i = 0.1$–$3.3 \times 10^{-2}$ and activation energy of Ea = 1.35–1.29 eV in the temperature range of 300–700 °C. It was revealed that an increase in the concentration of samarium oxide leads to an increase in ionic conductivity and a decrease in activation energy. A common feature of $CeO_2$-based solid solutions is a possibility for the interaction of oxygen vacancies with dopant cations as a result of lattice deformation at doping to yield local structures, requiring an additional energy for their release to participate in transfer processes. The bonding energy in these structures is determined by Coulomb interactions, also involving the component relating to the lattice relaxations concentrating near the defects and depending on the dopant nature. As a result of doping with $Sm_2O_3$, oxygen vacancies in a $CeO_2$-based distorted cubic lattice of the fluorite type tend to interact with the dopant cation yielding singly charged local defect structures of $(Sm'_{Ce} - V_O^{\bullet\bullet} - Sm'_{Ce})^{\bullet}$ and $(Sm'_{Ce} - V_O^{\bullet\bullet})$ [4,8], affecting conductivity. The temperature increase leads to the intensification of thermal vibrations in local structures, making oxygen vacancies more free and capable of diffusion, thus providing an increase in conductivity.

The considered method based on co-precipitation of hydroxides makes it possible to synthesize highly dispersed powders and dense, low-porous ceramics based thereon, featuring electrical conductivity that is about twice as high compared with similar compositions obtained by co-crystallization techniques [25]. The resulting ceramic nanomaterials can be used as effective bulk electrolytes for medium-temperature solid oxide fuel cells, since they meet all the requirements for electrolyte materials useful in alternative power sources of this kind.

**Author Contributions:** Conceptualization, O.A.S. and M.V.K.; methodology, M.V.K. and M.Y.A.; software, M.Y.A.; validation, O.A.S., M.V.K. and S.V.M.; formal analysis, S.V.M. and D.A.D.; investigation, D.A.D. and M.Y.A.; resources, O.A.S. and M.V.K.; data curation, M.V.K., D.A.D. and S.V.M.; writing—original draft preparation, M.V.K., S.V.M. and D.A.D.; writing—review and editing, M.V.K. and S.V.M.; visualization, M.Y.A. and D.A.D.; supervision, O.A.S. and M.V.K.; project administration, O.A.S.; funding acquisition, O.A.S. All authors have read and agreed to the published version of the manuscript.

**Funding:** The study is supported by the State Assignment for the Institute of Silicate Chemistry of the Russian Academy of Sciences (State registration numbers AAAA-A19-119022290091-8, 0081-2022-0006, 0081-2022-0007). The development of the methodology of DFT calculations was carried out as part of the research project "Chemistry, physics and biology of nanostate" (state registration number (TsIT and S): 0081-2022-0001).

**Institutional Review Board Statement:** Not applicable.

**Informed Consent Statement:** Not applicable.

**Data Availability Statement:** Not applicable.

**Acknowledgments:** The authors are thankful to the curator of the Engineering Center (Saint-Petersburg State Institute of Technology), D.P. Danilovich, for SEM characterization of the studied materials, and to the assistant professor of the Department of Theory of Materials Science (Saint-Petersburg State Institute of Technology), S.P. Bogdanov, for participation in the XRD studies.

**Conflicts of Interest:** The authors declare no conflict of interest.

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
