# Peer review of "Synthesis and Characterization of Ceria- and Samaria-Based Powders and Solid Electrolytes as Promising Components of Solid Oxide Fuel Cells"

_ceramics, doi:10.3390/ceramics5040078_

Round 1

Reviewer 1 Report

Having examined your manuscript titled "" I have seen that you  but a major revision is needed. 

1. Based on  XRD results, the authors concluded that annealing at 600 °C for 1 h results in highly dispersed  solid solutions with a cubic structure of the fluorite type with an average CSR size of ~7  nm.  The  crystallite size calculated does not seem correct according the XRD peaks present. Please add how you calculate the crystallite size and add as well some micrographs obtained by SEM or TEM to see the particle size of the powders obtained.

 At the further annealing at 600- 1,300 °C, the author claims that the powder is in the nanoscale. However, the XRD pattern at 1300 C show sharp peaks., it seems that you get a crystallite size in the nano range but probably the particles are in the micro range. Please add a micrograph by SEM of the powders. 

2. It is not clear  why the main idea is to obtain nanopowders if later you will consolidate the powders to get pellets.  What is the porosity of the pellets. 

Please include the SEM micrographs. 

When you talk about ceramic materials, pellets, with 65-69 nm size. This size is referring to crystallite size or grain size?  Please include the microstructural characterization of the consolidated pellets.

3. Do you consider low-porous ceramic materials when you got 91% relative density. Please comment on that. 

4. It is difficult to judge that you have working with nanomaterials. It seems that you have nanostructuration but you consolidate the powders.  Please explain how you obtain the values of open porosity of the ceramics.

Reviewer 2 Report

Reviewing your work, I found it interesting, however some points should be improved, and others clarified so that the work can be accepted in this famous and prestigious journal.

- The doping can improve several material properties and the choice for doping depends on several factors. With respect to this, the authors do not justify the doping (x = 0.05, 0.10, 0.20) concentrations used in this system. Due to the importance for this system, a reasoned explanation must be attached as part of the motivation and justification of the work.

- The purity of raw materials must be provided

-The quality of all figures should be improved

-The section:  Study of the synthesized xerogels thermolysis, is very descriptive. Please, enrich the discussion by comparing your results with the literature.

-  The section: Low-temperature nitrogen adsorption characterization of the microstructure of powders synthesized by co-precipitation of hydroxides, is weak; evidently there are visible changes in: Specific surface area and Average pore size, and they were not discussed heavily. How doping influences these textural parameters

-  The section: Study of the crystal structure of the obtained nanopowders; must be improved. A stronger discussion about structural changes should be made in terms of dopants and structural defects. The crystallographic card for identifying XRD patterns should be included, as well as Miller indices.

- In summary, it is appreciable that the authors study: Synthesis and Characterization of Ceria and Samaria Based Na  nopowders and Solid Electrolytes as Promising Components of  Solid Oxide Fuel Cells. However, the most analyzes done are highly based on hypothetical and qualitative consideration based and there is no in-depth scientific discussion. Authors must include sentences, with more relevant content

Round 2

Reviewer 1 Report

The authors have took in account the commentos proposed. 

Reviewer 2 Report

This New version of the manuscript may be considered for publication